# Impacts of a record-breaking storm on physical and biogeochemical regimes along a catchment-to-coast continuum

**Seán Kelly**[1,2]*, **Brian Doyle**[1,2], **Elvira de Eyto**[2], **Mary Dillane**[2], **Phil McGinnity**[2,3], **Russell Poole**[2], **Martin White**[4], **Eleanor Jennings**[1]

**1** Centre for Freshwater and Environmental Studies, Dundalk Institute of Technology, Louth, Ireland, **2** Marine Institute, Furnace, Newport, Mayo, Ireland, **3** School of Biological, Earth & Environmental Sciences, University College Cork, Cork, Ireland, **4** Earth & Ocean Science, National University of Ireland Galway, Galway, Ireland

* sean.kelly@dkit.ie

**Data Availability Statement:** All meteorological and hydrological data files used in the analyses presented in this study are available for download from: http://data.marine.ie/geonetwork/srv/eng/

## Abstract

The impacts of changes in climate are often most readily observed through the effects of extremes in local weather, effects that often propagate through multiple ecosystem levels. Precise effects of any extreme weather event depend not only on the type of event and its timing, but also on the ecosystem affected. Here the cascade of effects following the arrival of an atmospheric river (directed by record-breaking Storm Desmond) across terrestrial, freshwater and coastal zones is quantified, using the Burrishoole system on the Atlantic coast of Ireland as a natural observatory. We used a network of high-frequency *in-situ* sensors to capture in detail the effects of an unprecedented period of rainfall, high wind speeds and above-average winter air temperatures on catchment and estuarine dynamics. In the main freshwater lake, water clarity decreased and acidity increased during Storm Desmond. Surface heat input, due to a warm and moist above-lake air mass, was rapidly distributed throughout the water column. River discharge into the downstream coastal basin was estimated to be the highest on record (since 1976), increasing the buoyancy flux by an order of magnitude and doubling the water column stratification stability. Entrainment of salt into the outflowing freshwater plume exported resident salt from the inner estuarine basin, resulting in net salt loss. Here, the increased stratification markedly reinforced isolation of the bottom waters, promoting deoxygenation. Measurements of current between the inner estuarine basin and the adjacent coastal waters indicated a 20-fold increase in the volume of seaward flowing low-salinity water, as a result of storm rainfall over the watershed. Storm impacts spanned the full catchment-to-coast continuum and these results provide a glimpse into a potential future for hydrological systems where these severe hydroclimatic events are expected to occur more frequently.

catalog.search#/metadata/ie.marine.data:dataset.3979.

**Funding:** SK was funded under the Marine Research programme by the Irish Government (Cullen Fellowship CF/15/03 and BEYOND 2020 PBA/FS/16/02). PMcG was also supported by the Marine Institute under the Marine Research Programme (RESPI/FS/16/01). https://www.marine.ie/Home/home. The funders had no role in study design, data collection and analysis, decision to publish, or preparation of the manuscript.

**Competing interests:** The authors have declared that no competing interests exist.

# Introduction

The attribution of individual extreme weather events to directional climate change is highly complex, given natural climatic variability and the overall rarity of such events [1,2]. However the magnitude and frequency of extreme weather events are projected to increase given anthropogenically-influenced changes in atmospheric conditions [3]. Changes in long-term mean temperatures, for example, are also likely to be associated with changes in heat extremes (e.g. more severe heatwaves) [2,4]. For precipitation, an increase in the intensity and frequency of heavy rainfall and flood events is consistent with expectations given increasing atmospheric water vapour [5,6]. Relationships between observed trends in storm and cyclone activity and climate forcing are more nuanced and uncertain [5]. Evidence does suggest, however, that increased storm magnitude over the past number of decades is linked to warmer tropical sea surface temperatures [7] and model projections indicate that more hurricane-force storms may be experienced in western Europe due to rising sea surface temperatures in the Atlantic [8]. Therefore, while not singularly attributing such events to climate change, occurrences of extreme weather represent the most tangible way in which the public perceive and recognise climate change. These events thus represent important opportunities to motivate public concern for climate action and policy to mitigate continued carbon dioxide emissions [9,10]. Furthermore, quantifying the impacts of episodic weather events across ecosystems can provide a valuable insight into how these systems respond and can serve as a tool for predicting which specific geographical areas and ecosystems may be most vulnerable to increased frequency or intensity of different types of extreme weather events. However due to their short-lived and unpredictable nature, knowledge of the consequences of extreme events for physical, chemical or biological regimes is scarce and often very difficult to capture [11,12].

Freshwater systems (lakes and their surrounding catchments) represent ideal candidate systems for observing the effects of climate and extreme weather events and have developed a reputation as 'sentinels' of climate change [13,14]. This is due to their intrinsic relationship with the overlying atmosphere and surrounding landscape, with lakes integrating changes in meteorological (e.g. air temperature, wind dynamics, solar radiation, precipitation) and terrestrial (e.g. catchment run-off) conditions. Large magnitude, short-lived changes in any of these variables have the capacity to affect biotic and abiotic conditions in lakes, although the precise changes depend on the category and timing (season) of the event as well as lake-type (e.g. oligotrophic versus eutrophic, shallow versus deep, mixed versus stratified etc) [12]. For example, in thermally stratified lakes storm events may modify the thermal structure by destabilising the water column and deepening the biologically-productive surface layer [15–17]. Such changes may reduce water clarity (e.g. [11]) or cause exchange between epilimnetic and nutrient-rich, oxygen-poor hypolimnetic waters [18]. In contrast, in mixed lakes without significant vertical thermal gradients such events may have less of an impact on whole-lake mixing and thus ecological regimes. Heavy rainfall, by increasing rates of catchment run-off, can deliver particulate and dissolved organic matter from the surrounding catchment, which can increase rates of primary-producer respiration [19] and reduce light penetration [20]. The extent to which lake ecosystems can recover (resiliency) depends on the type of surrounding catchment area as well as current nutrient status of the lake, with clear-water, oligotrophic lakes perhaps likely to undergo more significant changes than eutrophic or humic lakes [11,17,19].

Estuarine environments also tend to respond rapidly to extreme weather events also, although in addition to being influenced by atmospheric and terrestrial conditions, estuaries integrate changes in the adjacent marine environment, making them one of the most sensitive habitats to extreme weather events and climate change in general [21]. How estuarine systems respond to extreme climate events varies between individual sites, with classification based on

geomorphology (depth, width), salinity structure (well-mixed through to stratified) and tidal range [22]. As with lakes, generalisation of the effects of extreme weather events on estuaries can also be difficult–here we consider the effects of storms and floods which subject estuarine systems to wind-forcing and large volumes of freshwater input.

Heavy rainfall events may influence salinity structure, with river runoff potentially pushing the salinity intrusion limit down-estuary, exporting nutrients and salt in the process [23,24] and generating stratification in shallow well-mixed estuaries [25]. In deeper, semi-enclosed lagoonal and fjordic estuaries, which are subject to relatively weaker tidal forcing, high river flow following precipitation could reinforce stratification, increasing renewal time of deep-water masses and prompting concern over deoxygenation [26]. If the same floods lead to an influx of organic matter and nutrients from runoff, heightened respiration may compound the risk of bottom water oxygen depletion and coastal water quality in general is likely to decline [27,28]. The way in which estuarine stratification dynamics, such as the baroclinic flow fields, adjust to large quantities of freshwater input is of fundamental importance to estuarine physics and salt balance regulation [29–31]. An added complexity to the stabilising effects of freshwater input is that strong winds are often associated with storm events and serve to destabilise the water column potentially mixing saltwater and freshwater layers. In estuaries with deoxygenated deep-water, such storms could enhance upwelling of suboxic waters to shallower zones inhabited by aerobic organisms [32].

The frequency and severity of storm category weather events during winter 2015/16 in Ireland and the Britain was considered exceptional, peaking with the extratropical cyclone Storm Desmond during 4–6 December [33]. Rainfall during Desmond broke the 24- and 48-hour British rainfall records and multiple rivers throughout Ireland and Britain recorded highest ever peak discharge [34]. Desmond was caused by enhanced horizontal water vapour transport from the Atlantic Ocean, with the plume of moist air generating an 'atmospheric river' and causing extreme precipitation following orographic lifting along mountainous western Irish and British coastlines [35]. The increased heat content of the North Atlantic has led to long-term increases in atmospheric humidity, which could cause more extreme precipitation events like Desmond to occur over Ireland, Britain and western Europe [35,36]. In this study, we used *in-situ* high frequency sensor equipment to investigate the impacts of this remarkable succession of winter storms and floods on the contiguous habitats of a catchment-lake-estuarine system. We also analysed the extent to which these episodic events modified the physical and biogeochemical regimes in the system and which ecosystems were most affected by this extreme winter flood. Our study site, the Burrishoole catchment, is representative of many upland peat catchments that occur throughout Ireland and the United Kingdom, particularly along western seaboards. Thus we envisage that the results from this analysis could be generalised to many similar systems that are also anticipated to face the same increased occurrence of extreme rainfall events.

## Methods

### Site description

The Burrishoole catchment (~100 km$^2$) is a predominantly upland blanket peat, oligotrophic catchment located within the Nephin Beg mountain range on the west Atlantic coast of Ireland (Fig 1). Rivers and streams are generally acidic or neutral and water chemistry depends on local and regional climate as well as surrounding soil composition in each of the sub-catchments [37]. Given its coastal proximity, the catchment generally experiences a temperate climate with mild winters (mean December-February 2005–18 air temperature of 6.0˚ C) and summers (mean June-August 2005–18 air temperature of 14.3˚ C) and a diurnal sea breeze

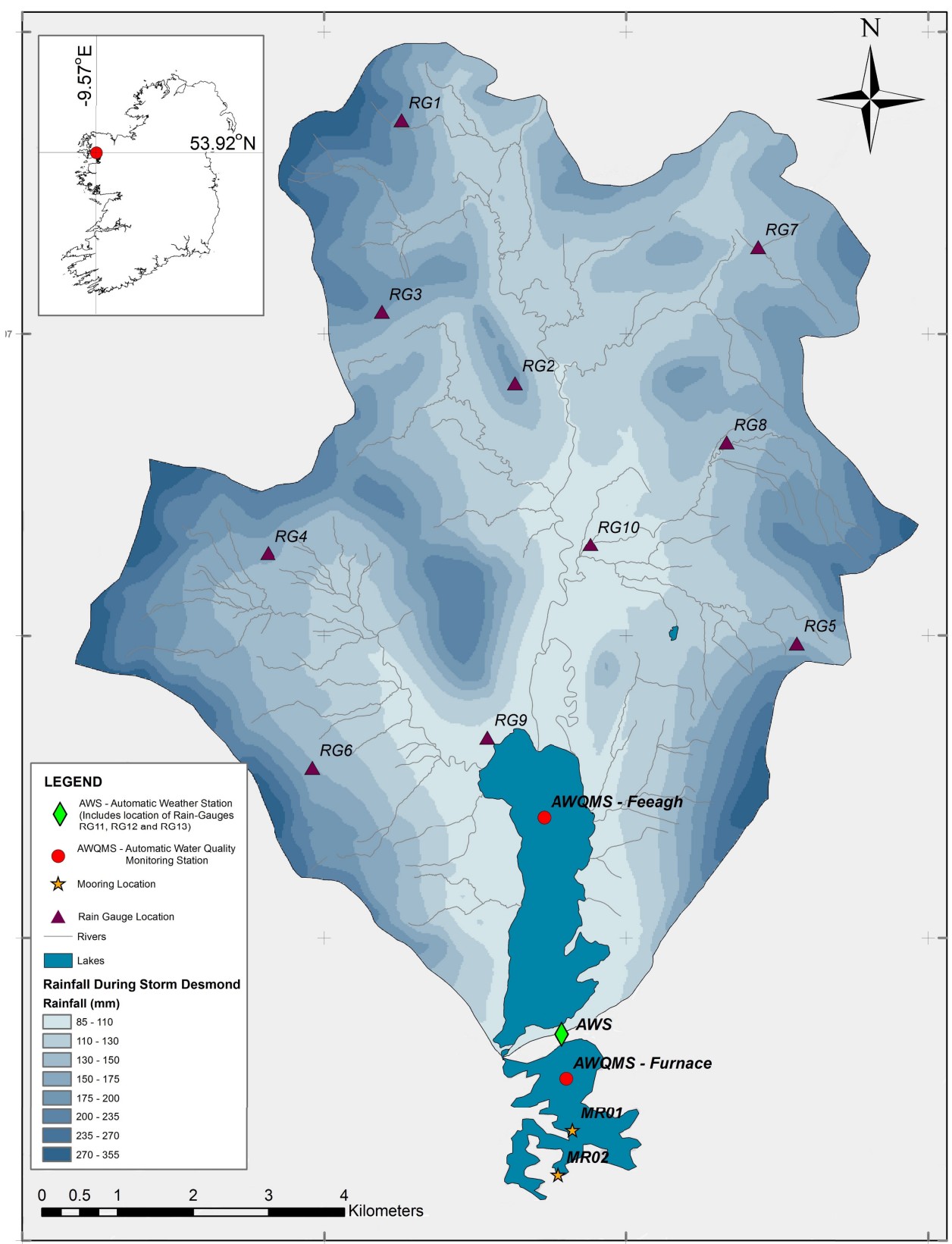

**Fig 1. Spatial distribution of rainfall over Burrishoole catchment during storm Desmond.** Measured by 13 rain gauges (RG) (note that RG 11, 12 and 13 are situated in the vicinity of the Automatic Weather Station (AWS)). Also shown are the locations of the Automatic Water Quality Monitoring Stations (AWQMS) on Lough Feeagh and Lough Furnace and the additional instrumented moorings MR01 and MR02 on Furnace, described in the text. Shapefiles used for lakes and rivers shown here are available at: http://gis.epa.ie/GetData/Download.

with mean wind speeds of ~5 m s$^{-1}$ [38]. Burrishoole's primary lake basin is Lough Feeagh, with a surface area of 3.95 km$^2$, max depth of 46 m, mean depth of 14.5 m and volume of 5.9 x 10$^7$ m$^3$. The water of Feeagh is oligotrophic and humic, with a mean secchi disk depth of 1.7 m, pH of 6.6 and annual mean total nitrogen and phosphorus of 430 µg l$^{-1}$ and 6.1 µg l$^{-1}$ respectively [39]. Feeagh is located at the base of the Burrishoole catchment, draining a ~84 km$^2$ watershed into Lough Furnace through two short outflows, the Salmon Leap and (man-made) Mill Race rivers. Furnace is a deep lagoonal estuary (M$_2$ semi-diurnal tide dominant) with a surface area of 1.5 km$^2$, max depth of 21 m and mean depth of 7.9 m. The system comprises a main inner basin (volume of 6.75 x 10$^6$ m$^3$) connected to the open coastal waters of Clew Bay by a 1-km-long shallow and narrow channel. Tidal currents transporting coastal water must traverse this channel and two narrow topographic constrictions before reaching the inner basin [38]. The inner Furnace basin is notable for its strong saline stratification and deep anoxia, resembling a meromictic saline lake, although ventilation of the bottom water by dense tidal inflows occurs irregularly [38]. A research station operated by the Marine Institute is situated on the northern shore of Furnace and has served primarily as an international index site for migratory fish populations (Atlantic salmon (*Salmo salar* L.), trout (*Salmo trutta* L). and European eel (*Anguilla Anguilla* L.)) in the North Atlantic for half a century [40,41]. The upper parts of the catchment have Special Area of Conservation (SAC) status for the conservation of Atlantic salmon and otter (*Lutra lutra*), two species listed in Annex II of the EU Habitats Directive. Lough Furnace is part of the Clew Bay Complex Special Area of Conservation (SAC site code 001482).

## Data collection and processing

This study relied on data sources routinely collected as part of the Marine Institute's long-term environmental monitoring programme in Burrishoole. Spatial extent of rainfall over the entire Burrishoole catchment was recorded by 13 tipping bucket rain gauges (Davis Instruments Corp, Hayward, CA, USA) located throughout the catchment and lake and lagoon levels were monitored by pressure sensors (OTT, Kempton, Germany) recording every 15 minutes (Fig 1). Automatic water quality monitoring stations (AWQMS) were located on Feeagh and Furnace (Fig 1). Each AWQMS was instrumented with on-lake meteorological arrays recording wind speed and direction, air temperature, shortwave radiation and surface pressure at 2-minute intervals. A multi-parameter sonde (Hydrolab DS5, OTT, Kempton, Germany) was suspended from each AWQMS; on Feeagh this sonde was kept at 0.9 m below the surface recording every 2 minutes whilst on Furnace the sonde was attached to an undulating winch profiler, recording 4 full water column profiles daily (at 00:00, 06:00, 12:00 and 18:00 hours). Parameters measured on each sonde included temperature, conductivity, pH and dissolved oxygen. The Feeagh AWQMS also contained a surface nephelometer (Chelsea Technologies Group Ltd, Surrey, UK) and thermistors (Labfacility Ltd., Bognor Regis, UK) that spanned the full water column at depth intervals of 2.5, 5, 8, 11, 14, 16, 18, 20, 22, 27, 32, 42 m, recording every 2 minutes. An additional automatic weather station (AWS) maintained by Met Éireann since 2005 was situated on the shoreline between both loughs, which recorded relative humidity and rainfall in addition to the same over-water meteorological variables recorded at each AWQMS (Fig 1).

On Furnace two additional instrumented moorings were located at the entrance channel into the main inner basin (Fig 1). MR01 (S1 Fig) contained a bottom-mounted, upward look-ing 1-MHz Nortek Aquadopp three-beam current profiler (ADCP) (Nortek AS, Rud, Nor-way). The ADCP sensor head, with a 25˚ beam angle to the vertical, was mounted at 0.2 m above the bed. The ADCP recorded in 0.5 m bin intervals, with a 0.4 m blanking distance and a velocity measure averaged over a 60 second ping interval. Two temperature and conductivity sensors (T/C) were attached to the same mooring (SBE-37 MicroCATs (Sea-Bird Scientific, Bellevue, Washington, USA)). One T/C sensor was mounted on the bottom ADCP frame and the second positioned ~1 m below the surface on an L-shaped mooring. An additional T/C, also recording pressure, was located further downstream (MR02, Fig 1). All of these instru-ments had synchronised internal clocks and logged measurements every 30 minutes from 13:30:00 November 26[th] 2015 until 11:30:00 January 21[st] 2016.

CTD downcasts from the Furnace AWQMS were extrapolated onto a standard depth grid spanning the water column from 0.6 m– 12.9 m in 0.15 m depth increments. Practical salinity (derived from actual conductivity and temperature) and water density (derived from salinity and temperature) were computed according to the UNESCO algorithm [42] for all sensors recording temperature and conductivity.

ADCP velocity vectors were low-pass filtered with a 6 hour cut-off frequency to remove higher frequency tidal harmonics. Low-passed horizontal velocity vectors were then rotated into along- (v) and across-channel (u) components using principal components analysis (PCA) [43]. Whilst the standard ADCP bin sizes spanned 0.5 m in the vertical, the lowest bin was weighted to account for the bottom 1.1 m of the channel (i.e. 0.5 m bin size + 0.4 m blank-ing distance + 0.2 m frame height above the bed). For each individual ADCP profile, the uppermost usable bin was defined as the shallowest bin below the computed depth of sidelobe interference with contaminated shallower measurements discarded. In order to account for this portion of the upper water column not measured (~10–15%), each current profile was extrapolated to the surface level using a constant velocity from the uppermost usable bin.

## Data analysis

Precipitation recorded by each of the 13 rain gauges throughout the catchment were summed to 24 hour totals. To estimate the total rainfall volume that fell in the catchment within a spe-cific time period, the raster calculator routine in the 'Spatial Analyst' toolbox in the GIS Soft-ware ArcMap 10 [44] was used. The raster calculator routine allows mathematical calculations to be conducted on existing raster data sets, in this case a raster map of the topography of the catchment. The topographic raster data comprised a 10 by 10 metre grid of the catchment with an altitude above sea level value at the centre of each grid square. To estimate rainfall volumes during Storm Desmond, an altitude-rainfall regression equation was calculated for the time period 4–6 December 2015. The rainfall volumes for each grid square were summed to get the total rainfall volume. However, the altitude-rainfall volume relationship during storm Des-mond ($R^2$ = 0.32) was not strong when compared to a 14-year study period, 2004–2017 ($R^2$ = 0.82). Analysis of rainfall volumes during the storm showed that lower volumes of rainfall were recorded in the rain gauges at higher altitudes in the catchment than would otherwise be expected. Presumably the breakdown of the altitude-rainfall volume relationship at higher ele-vations in the catchment is a result of high wind speeds causing 'under-catch' of rainwater dur-ing the storm. Under-catch is a common phenomenon affecting the accuracy of rain gauges, where high wind speeds during storms cause wind flow deflections and turbulence patterns around the gauge funnel which in turn cause some of the raindrops to be ejected from the fun-nel system [45].

To correct for this, the rainfall volumes for the three rain gauges at the greatest altitudes were revised. Using the AWS rain gauge (Fig 1A) as an index site, the rainfall volume measured at this site during Storm Desmond was used to estimate the rain volumes at the three highest altitude sites. Coefficients between the index site and each of the high altitude sites were calculated over the 14 year period for which rain gauge data was available. The rainfall volume at AWS was multiplied by each of the rainfall volume coefficients for the three highest altitude sites to adjust for under-catch. Following adjustment, the altitude-rainfall volume relationship during the specific Storm Desmond timeframe improved to give an $R^2$ value of 0.73 and a revised total rainfall during Storm Desmond was estimated using this method.

Storm effects on dynamical transfer processes of heat and momentum across the air-water interface on both Feeagh and Furnace were assessed by calculating the net surface heat flux and energy input due to wind stress. The surface heat flux ($Q_{shf}$, W m$^{-2}$) over the month of December 2015 was calculated following [46]:

$$Q_{shf} = Q_{swin} + Q_{lwin} + Q_{lwout} + Q_h + Q_e \qquad (1)$$

where $Q_{swin}$ (W m$^{-2}$) is the net incoming shortwave solar radiation (accounting for reflected shortwave due to surface albedo) and $Q_{lwin}$ and $Q_{lwout}$ are the incoming and outgoing thermal radiation (W m$^{-2}$). $Q_h$ (W m$^{-2}$) is the sensible heat flux, $Q_h = \rho_a C_{Pa} C_H W_z (T_s - T_z)$, and $Q_e$ (W m$^{-2}$) is latent heat flux, $Q_e = \rho_a L_v C_E W_z (q_s - q_z)$, where $\rho_a$ is the density of air at the air-sea interface (kg m$^{-3}$), $C_{Pa}$ is the specific heat of air (J kg$^{-1}$°C$^{-1}$), $C_H$ the turbulent transfer coefficient for heat, $W_z$ the wind speed at height $Z$ above the water (m s$^{-1}$), $T_s$ the surface water temperature (°C), $T_z$ the air temperature at height $Z$ above the water surface (°C), $L_v$ the latent heat of vaporization (J kg$^{-1}$), $C_E$ is the turbulent transfer coefficient for humidity, $q_s$ the saturated specific humidity (kg kg$^{-1}$) at $T_s$ and $q_z$ the specific humidity of unsaturated air (kg kg$^{-1}$) at height $Z$ above the water surface. One caveat relating to the heat flux calculations presented herein is that the humidity measurements were only available from the shoreline meteorological station and not directly above either Feeagh or Furnace water surface. Positive values of each variable in Eq (1) indicate downward flux of heat; negative values indicate loss of heat to the atmosphere.

Mechanical energy transfer from the wind to the water was parameterised as the rate of work per unit area by wind stress at 10m above the water surface (W m$^{-2}$) [47]:

$$E_{10} = \rho_a C_D W_{10}^3 \qquad (2)$$

where $C_D$ is the surface drag coefficient. Turbulent transfer coefficients for heat ($C_H$), momentum ($C_D$) and humidity ($C_E$) were scaled to 10m above the water surface by correcting for atmospheric stability [46,48].

An informative measure for saline Furnace was to estimate a buoyancy flux associated with the storm through the surface, $B_f$ (m$^2$ s$^{-3}$), including a moisture flux term (e.g. [49]):

$$B_f = g \left( \frac{\alpha}{\rho_0 C_P} Q_{shf} + \beta S_0 (\frac{Q_f}{A_s} + P - E) \right) \qquad (3)$$

where $g$ is gravity (m s$^{-2}$), $\rho_0$ is surface water density (kg m$^{-3}$), $C_P$ the specific heat capacity of water (J kg$^{-1}$ K$^{-1}$), $\alpha$ and $\beta$ are the thermal expansion ($\alpha = \rho_0^{-1} \partial \rho / \partial T$)) and haline contraction ($\beta = \rho_0^{-1} \partial \rho / \partial S$) coefficients, $S_0$ is surface salinity and freshwater flux (m$^3$ s$^{-1}$) is given by river discharge ($Q_f$) plus direct precipitation ($P$) minus evaporation ($E$) over the surface area of the lough ($A_s$, m$^2$). Positive $B_F$ indicates buoyancy gained at the water surface.

How storm events influenced full water column stability in each water body was assessed by calculating the Schmidt stability index ($S_s$, J m$^{-2}$) [50]:

$$S_s = gA_s^{-1} \int_0^{Z_{max}} (z - \bar{z})(\rho(z) - \bar{\rho})A(z)dz \tag{4}$$

where $\bar{z}$ is the depth (m) at which mean density occurs, $\rho(z)$ is density at depth $z$, $\bar{\rho}$ is volume weighted mean density and $A(z)$ is basin area at depth $z$. $S_s$ estimates the amount energy needed to fully mix the water column, providing a good quantitative measure of stratification and destratification processes.

In Feeagh, which is destratified by December, the total lake heat content ($H_{tot}$, J m$^{-2}$) was identified as a potentially more informative measure of internal lake response to atmospheric forcing compared to $S_s$:

$$H_{tot} = \rho_0 C_P A_s^{-1} \int_0^{Z_{max}} T(z)A(z)dz \tag{5}$$

In Furnace, storm winds would be anticipated to destabilise the stratified water column yet the large volumes of freshwater discharge would be anticipated to reinforce vertical stratification through a positive buoyancy input. An appropriate parameterisation of the wind-driven response of the semi-enclosed (i.e. finite-dimension), 2-layered Furnace basin during the storms is the dimensionless Wedderburn number ($W_b$) (e.g. [51]):

$$W_b = \frac{R_i}{(l'/h)} \tag{6}$$

where $R_i$ is the surface-layer Richardson number $R_i = g'h/u_*^2$, with $g'$ the reduced acceleration of gravity across the pycnocline ($g' = g(\Delta\rho/\rho_0)$), $h$ the depth of the surface layer (m) and $u_*$ the surface water shear velocity (m s$^{-1}$) given as $u_* = \left(\left(\frac{\rho_a}{\rho_0}\right)C_D W_{10}^2\right)^{0.5}$. Wind speeds were low-pass filtered with a cut-off frequency of ¼ the fundamental mode internal wave period [32]. $l'/h$ is the aspect ratio of the basin, with the effective length scale $l'$ defined here as the length of the unbroken pycnocline along the NE-SW axis, the direction of wind stress during Desmond.

In general, large $W_b$ values ($> 10$) imply a sharp, flat interface is maintained; smaller values ($1 < W_b < 10$) indicate partial upwelling of the pycnocline at the upwind basin end and subsequent internal seiching following wind cessation; $W_b \approx 1$ implies full pycnocline upwelling whilst still smaller values (e.g. $< 0.5$) indicate full surfacing of sub-pycnocline waters with formation of longitudinal density gradients and ultimately basin-scale overturns [51–53].

Estuarine exchange flow through the connecting channel between Furnace and Clew Bay controls transport dynamics in and out of the main inner Furnace basin and links the Burrishoole watershed to the coastal ocean. Modification of salt content of dense bottom inflows due to mixing with lower salinity surface outflows regulates the residence time of saline water masses in the inner basin and thus the oxygen content of deep waters [38]. The impact of the storms on diapycnal mixing in this region was investigated using ADCP profiles (MR01 Fig 1), with squared shear computed as:

$$S^2 = \left(\frac{\partial u}{\partial z}\right)^2 + \left(\frac{\partial v}{\partial z}\right)^2 \tag{7}$$

where z is depth in the channel. As only two measures of density were available at the surface and bottom near the ADCP, we computed a full water column 'bulk' buoyancy frequency N

as:

$$N = \left[ -\left(\frac{g}{\rho_0}\right)\left(\frac{\partial \rho}{\partial z}\right) \right]^{0.5} \tag{8}$$

$S^2$ was averaged over the full water column and the gradient Richardson number calculated as (e.g. [54]):

$$Ri_g = \frac{N^2}{S^2} \tag{9}$$

It was envisaged that $Ri_g$ would provide a simple diagnosis of the dynamic stability of flow through this important section of the system, integrating the effects of wind, river flow and tidal currents on shear and stratification. $Ri_g > 0.25$ is one criterion for the occurrence of flow instabilities [54].

To compute volume and salt fluxes in and out of the inner Lough Furnace basin, the full entrance cross-section at MR01 was divided into subsections in order to overcome the limited spatial coverage of the sensors (S1 Fig). An echosounder transect provided accurate bathymetry. Firstly, the entranceway was divided laterally into a deeper trough section and shallower section to the east. The deeper channel region was divided vertically in 3 sections, with section averaged salinity in the uppermost and lowermost sections set equal to the values recorded by the T/C sensors; an interpolated salinity value was used for the intermediate section. A corresponding set of along-channel, area-averaged velocity vectors were computed from the ADCP profiles. To account for the unmonitored shallower (~2m) east section we used a simplistic approach and extrapolated across the salinity and velocity from the uppermost section in the deep channel, with the caveat that the resulting volume and salt flux values represented a first-order estimate.

Thus, within each subsection $i$, a timeseries of area $A_i$, along-channel velocity $v_i$ and salinity $S_i$ were derived. The net salt flux ($S_f$) was computed as the sum of flow rate and salinity through each area of the cross-section:

$$S_f = \langle \textstyle\sum_i v_i S_i A_i \rangle \tag{10}$$

$A_i$ for the top section in the deep channel and for the shallow side section increased or decreased with changing water level as measured by the ADCP pressure sensor. Angled brackets indicate averaging over a complete $M_2$ tidal cycle. Positive values indicate fluxes into the inner Furnace basin and negative values indicate fluxes out (seaward). Volume fluxes were estimated by omitting the salinity term in Eq (10).

## Results

Using the standard and revised rainfall-altitude relationships for the catchment rain gauges described in the Methods section, the total estimated rainfall volume over the Burrishoole watershed during Storm Desmond was in the range $12 \times 10^6$ - $14 \times 10^6$ m$^3$ (Fig 1). In terms of these rainfall volumes, Storm Desmond was a true record-breaker. Six out of thirteen rain-gauges recorded their historical maximum cumulative daily rainfall total during one day of the storm (5th December); the greatest daily rainfall total (135.9 mm d$^{-1}$) of any of these rain gauges was recorded at RG7 (Fig 1). For context, the mean daily rainfall total at RG7 during December 2004–2017 was 7.5 mm d$^{-1}$ and the historical maximum daily total (excluding December 2015) was 59.4 mm d$^{-1}$.

On Lough Feeagh the two single highest mean daily lake levels recorded since records began in 1976 occurred in November and December 2015, just 3 weeks apart (1.65 m on 15th

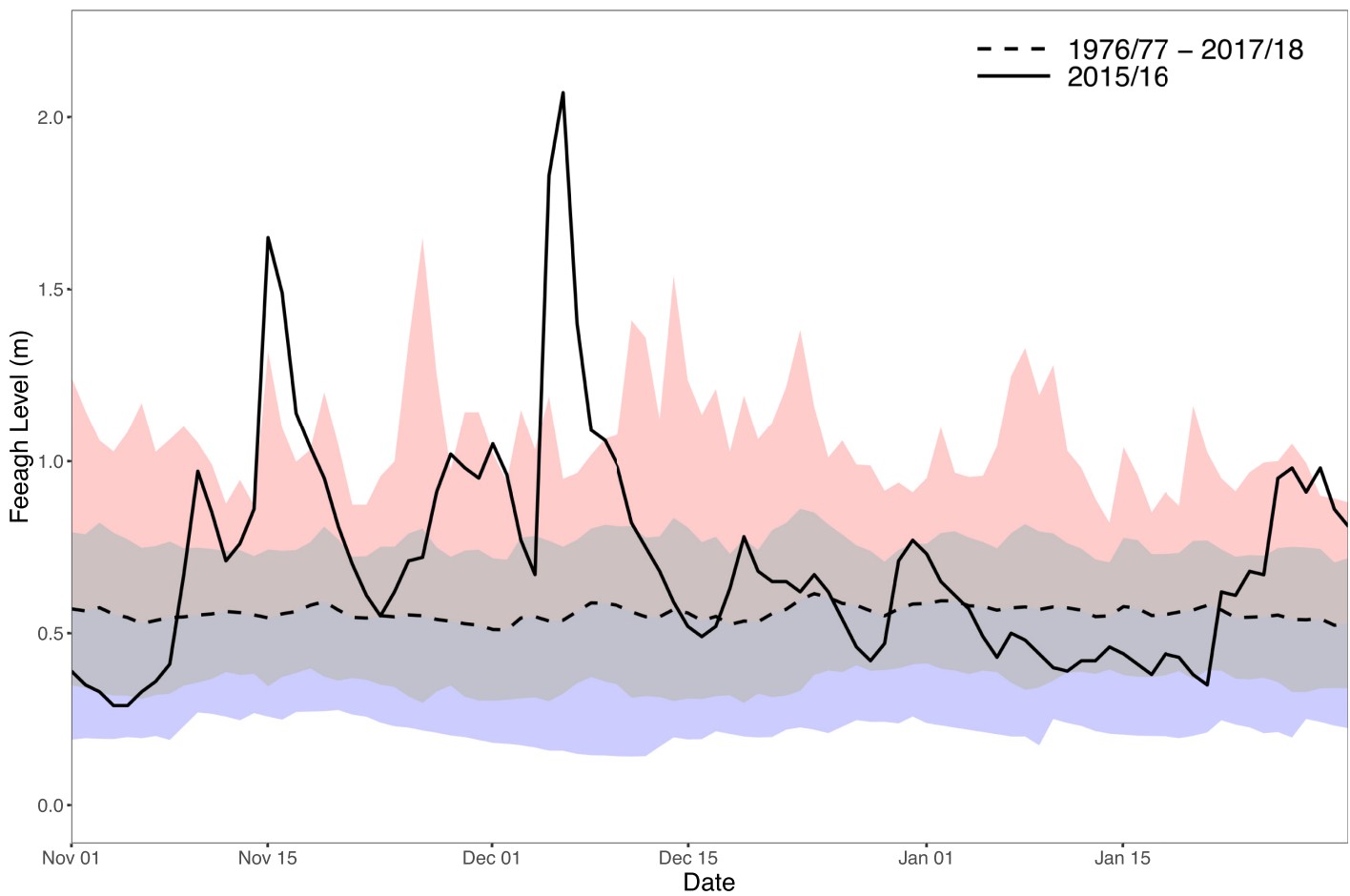

**Fig 2. Surface water levels (m) on Lough Feeagh.** Solid black line is daily mean November-January 2015/16; dashed black line is daily mean November-January 1976/77-2018/19 (grey shading is standard deviation around the mean). Red shaded area is historic max daily mean 1976–2018 and blue shaded area is historic min daily mean 1976–2018.

November and 2.07 m on 6th December (Fig 2)). It should be noted that during Desmond, the Lough Feeagh basin filled beyond carrying capacity and spilled over-land into downstream Furnace. This flood caused extensive structural damage to the Marine Institute's aquaculture research and migratory fish trapping facilities [55]. In addition to intense precipitation, a comparison of the mean December air temperature and wind speeds recorded at AWS (Fig 1) to monthly December averages over the period 2005–2018 revealed that mean air temperature (8.2°C 2015; 6.5°C 2005–2018) and wind speed (7.1 m s$^{-1}$ 2015; 5.6 m s$^{-1}$ 2005–2018) were both the higher than average. Following Desmond, two additional storm category weather events occurred on the 23rd ('Eva') and 29th ('Frank') December.

Persistent heavy rainfall throughout November ensured that catchment soils would have already been saturated when Desmond occurred; with very little storage capacity the rainfall over the 4–6 December would have followed overland routes, resulting in the drastic rise in water levels (Fig 2 and Fig 3A). Turbidity increased and pH decreased at the surface of Feeagh in the immediate aftermath of Desmond on 6th December (Fig 3B). In addition, total lake heat content increased by ~30 MJ m$^{-2}$, despite the overall seasonal decrease in water temperatures

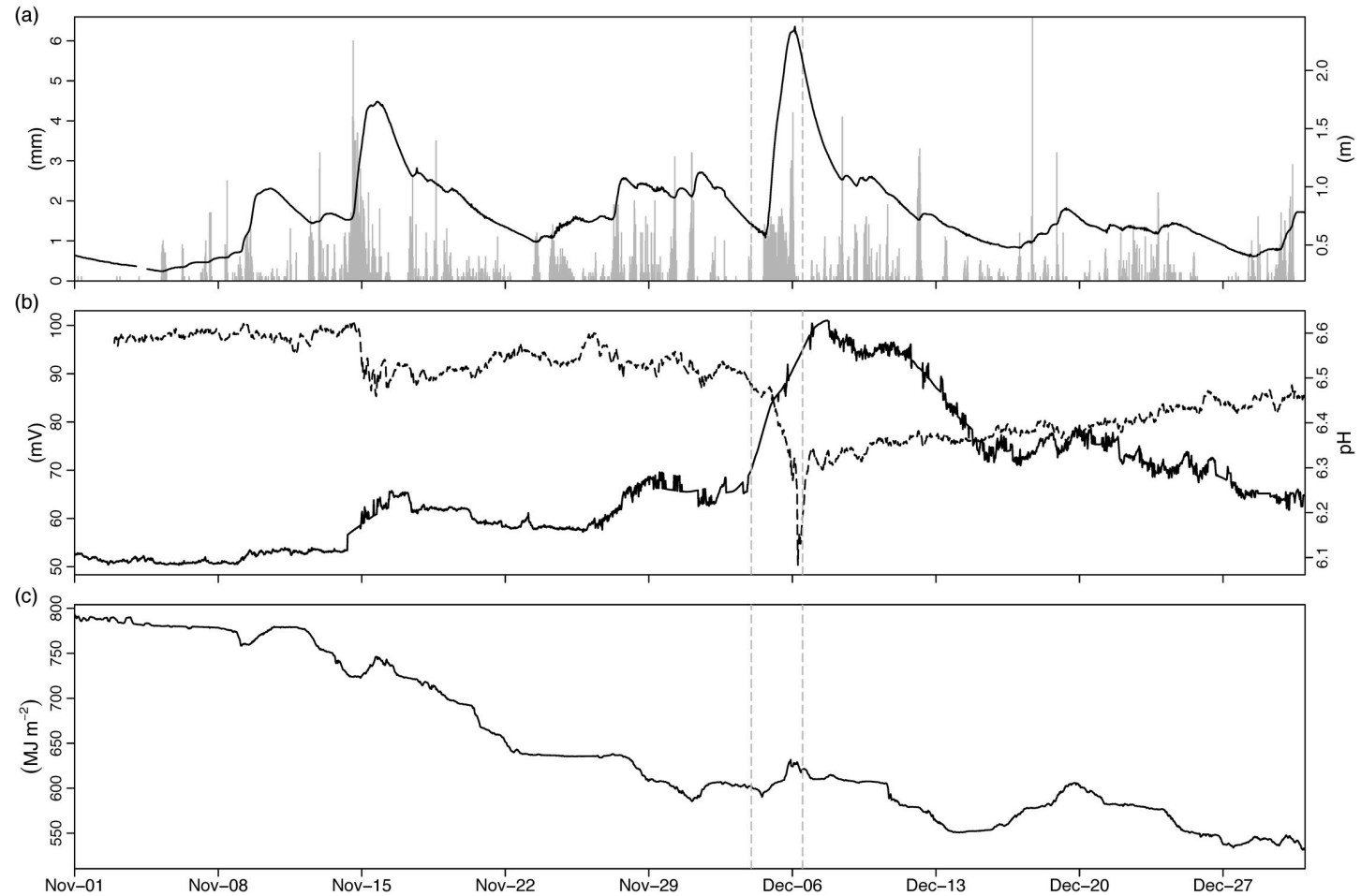

**Fig 3. 30 minute timeseries showing storm impacts on Lough Feeagh.** (A) rainfall (mm) recorded at AWS (grey bars) and Feeagh surface lake level (m) (B) surface pH (dashed line) and turbidity (mV) on Feeagh, (C) total heat content (MJ m$^{-2}$) of full Feeagh water column. Dashed vertical lines denote Storm Desmond.

from November onward (Fig 3C, black line). $S_s$ values were very low during this winter period, as the Feeagh water column is generally isothermal and fully-mixed.

Analysis of air-water transfer processes on Feeagh confirmed that a net positive transfer of heat (typically ~100–200 W m$^{-2}$) occurred during Desmond and other high wind events in December (Fig 4). Heat input from solar radiation was very slight and short-lived given the cloudy sky conditions and short day length during this time of year (Fig 4A). More important drivers of heat flux into the water appeared to be a slight net positive heating by thermal radiation (Fig 4B), an increase in sensitive heat flux due to increased over-lake air temperature (Fig 4C) and an increase in latent heat flux due to condensation of atmospheric water vapour at the surface of Feeagh (Fig 4E).

On Furnace the effects of large volumes of rainfall in the winter of 2015 were apparent as early as November, with a notable deepening of the halocline (red line, Fig 5A). Between 01 November and 06 December, the halocline had deepened by ~6 m (over 25% of the total water depth). Crucially, this haline stratification influenced the vertical distribution of temperature and dissolved oxygen (Fig 5B and 5C). The warm, anoxic sub-halocline basin water was completely isolated from the upper freshwater layer following Desmond, due to increased sharpness of oxygen and (inverse) thermal gradients. The strength of the stratification stability,

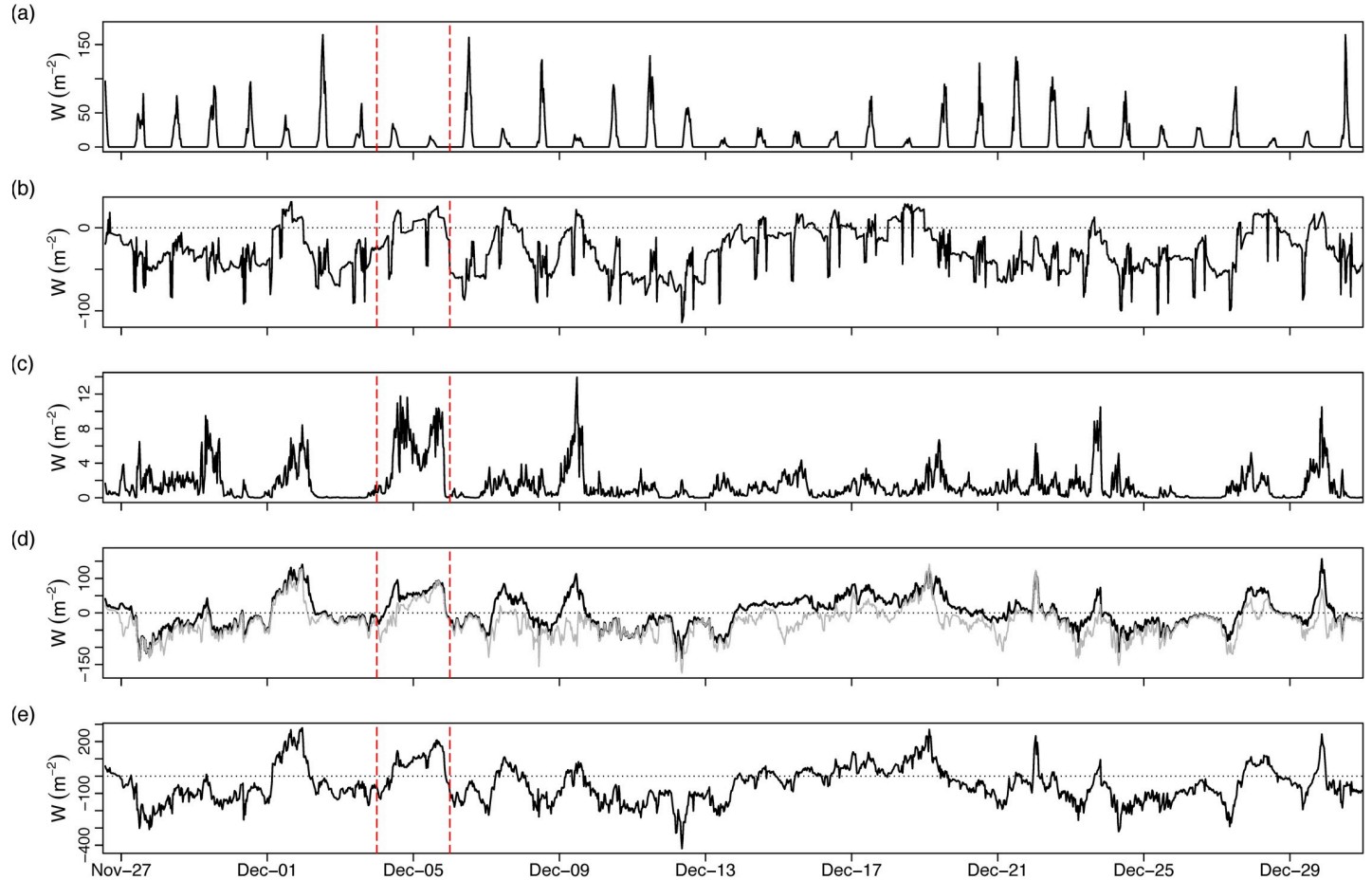

**Fig 4. 30 minute timeseries of atmosphere-water interaction on Feeagh during winter storms 2015.** (A) solar radiation (W m$^{-2}$), (B) thermal radiation (W m$^{-2}$), (C) energy input from wind stress (W m$^{-2}$), (D) latent (grey line) and sensible (black line) heat flux (W m$^{-2}$), (E) total surface heat flux (W m$^{-2}$). Positive values denote downward flux into the lake. Dashed vertical lines denote Storm Desmond.

as indicated by $S_s$ values, more than doubled over the course of the winter floods (~800 J m$^{-2}$ to ~2000 J m$^{-2}$) (Fig 5D).

The large quantities of rainfall leading up to and including Desmond culminated in a buoyant surface plume that increased the buoyancy flux into Furnace by an order of magnitude (Fig 6A). The stabilising effect of this buoyancy input increased the baroclinic restoring force and despite large wind stress values being reached during and after Desmond (Fig 6B), only partial upwelling occurred as indicated by non-critical (*i.e.* > 1) values of the inverse $W_b$ number (Fig 6C). The upper and lower dashed lines in Fig 6C represent a range of values over which partial upwelling is expected (equivalent to standard $W_b$ values in the range 3–6). On only one occasion did $W_b$ indicate significant upwelling of the pycnocline toward the surface (in the midst of Desmond) and on several occasions during particularly strong winds smaller upwelling events occurred. Inverse $W_b$ values falling in this range are indicative of subsequent internal seiche activity [53].

Timeseries of the instruments moored at MR01 and MR02 provided valuable insight into how the storm-related floods modified estuarine hydrodynamics at the entranceway to Furnace inner basin (Fig 7). Firstly, along-channel currents showed atypical dynamics during Desmond with strong seaward flow (negative values, denoted by blue) occupying the full extent of

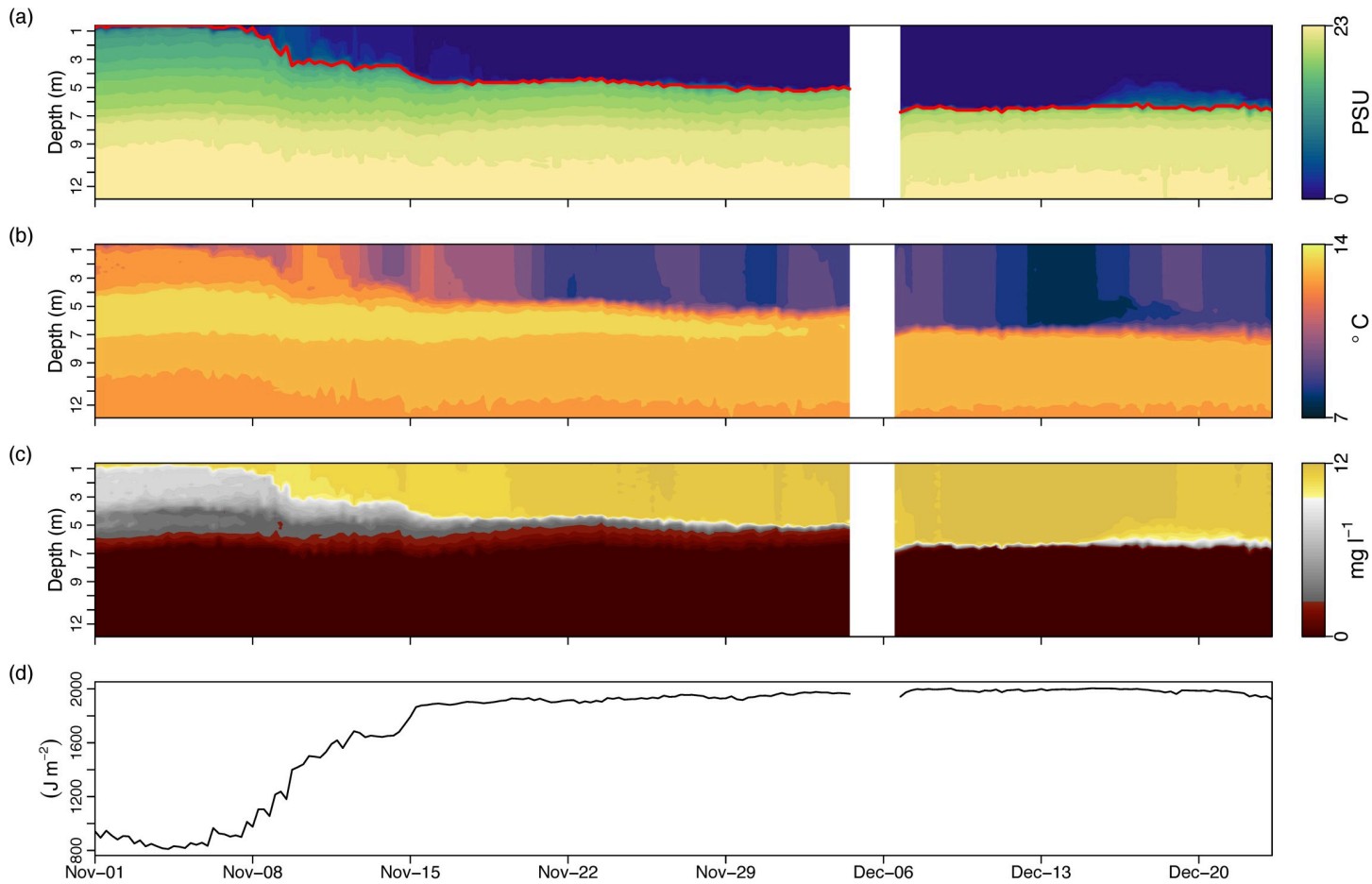

**Fig 5. Water column profiles (taken every 6 hours) on Lough Furnace pre- and post-Storm Desmond.** (A) salinity, (B) temperature (°C), (C) dissolved oxygen (mg l⁻¹), (D) Schmidt stability (J m⁻²). Note that the automated profiler malfunctioned during Storm Desmond (white gap, 4–6 December 2015).

the channel depth (Fig 7A). It should be noted that Desmond occurred during neap tides, meaning that only weak inflows (positive values, denoted by red) to Furnace would typically occur at this time. (Spring-neap tidal phase is indicated by the water level (black line, Fig 7A) as well as the salinity timeseries at MR02 (black line, Fig 7B)). Water level in the channel rose in the immediate aftermath of Desmond and fell again quickly, as the initial pulse of freshwater exited the system (Fig 7A). Surface (blue line), bottom (red line) and even lower estuary (black line) water was essentially comprised of freshwater from the end of November until spring tides on the 11th December, when saltwater inflows were observed along the channel bottom (Fig 7A and 7B). Curiously, a slight increase in surface and bottom salinity was detectable in the midst of Desmond (Fig 7B). Salinity at the surface remained almost at 0 for the remainder of the month and into January 2016, whilst bottom salinities rose gradually on each successive spring tidal phase causing restratification to occur. Aside from clear bottom inflow during spring tides, current profiles revealed that the flow direction in the upper water column (< 2 m from the surface) was dominated by along-channel wind direction (Fig 7A and 7C). The predominant wind direction during December was up-estuary, which opposed the outflowing flood water direction, causing surface flow reversals. Up-estuary winds forced outflow at intermediate depths and when this occurred during spring tides, a 3-layered exchange flow was

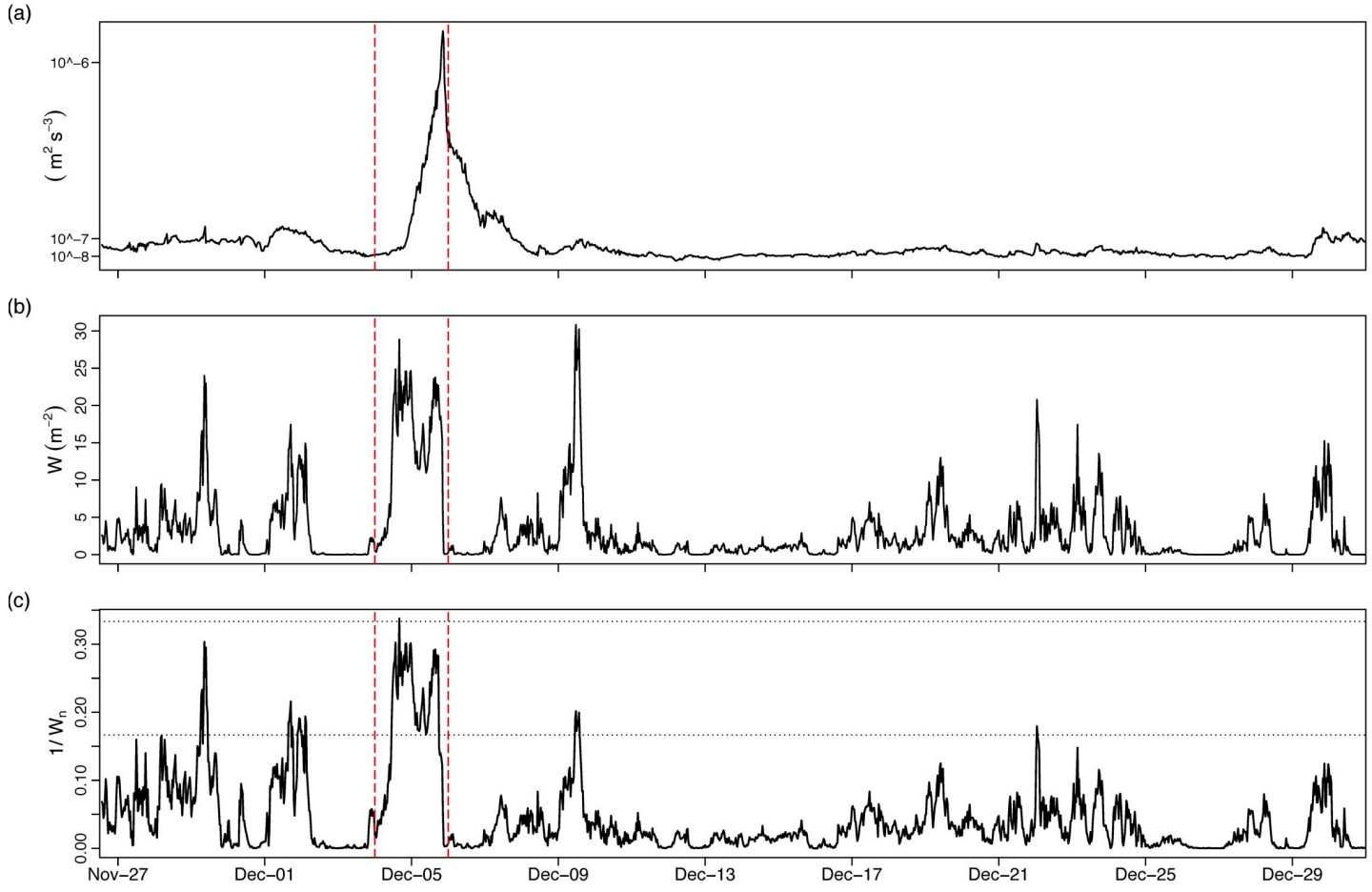

**Fig 6. 30 minute timeseries showing rain and wind impacts on Furnace.** (A) surface buoyancy flux (m$^2$ s$^{-3}$) (positive values indicate buoyancy input at surface), (B) energy input from wind stress (W m$^{-2}$), (C) inverse Wedderburn number (horizontal dashed lines denote partial and full upwelling limits). Dashed vertical lines denote Storm Desmond.

observed with bottom and surface inflow and intermediate outflow (*e.g.* December 11–18; December 25-January 01).

Intense shear was observed near the surface during strong wind events throughout December and January (Fig 7D). However in the lead up to and immediate aftermath of Desmond, only very weak and sporadic shear was observed further down in the water column. It was not until the restratification associated with the subsidence of the Desmond flood waters and the onset of spring tides (Fig 7E) that occasions of intense shear were supported near the channel bottom (*e.g.* ~December 18[th], ~Jan 5[th], Fig 7D). Analysis of S$^2$ and bulk water column $N^2$ (Fig 7E) and $Ri_g$ (Fig 7F) provided insight into relative contributions to flow stability during and after the floods. During the flood, stratification was absent and flow was completely turbulent through the entranceway, which had not been observed previously in this section [77]. However during the subsequent restratification, flow went through phases of stability and instability denoted by $Ri_g$ falling below 0.25 (dashed horizontal line). Instability was generally related to strong winds and tidal currents and was most critical just prior to 25[th] December, when stratification had weakened during neap tides. As the stratification continued to increase post-flood with successive spring tides, flow became increasingly stable and in January, despite occasions of wind- and tidally-driven shear, $Ri_g$ values remained above 0.25.

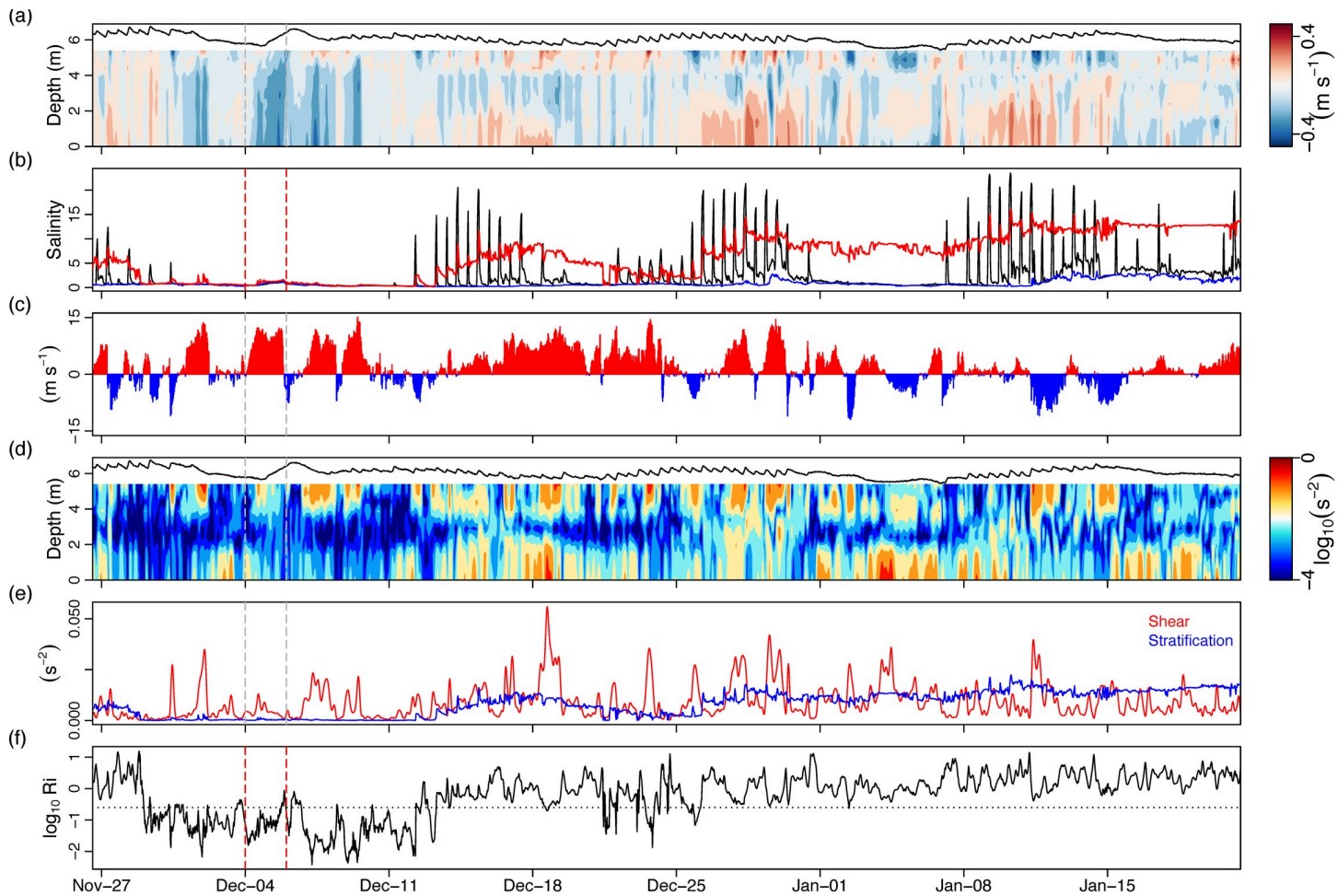

**Fig 7. 30 minute timeseries showing storm impacts on estuarine hydrodynamics.** (A) along-channel (350˚T) current velocities (m s$^{-1}$) at MR01 (black line is water surface; positive values indicate flow into Furnace inner basin, negative values indicate seaward flow), (B) surface (blue) and bottom (red) salinities at MR01, bottom salinity (black) at MR02, (C) along-channel winds (positive values indicate winds blowing toward Furnace inner basin), (D) squared shear (s$^{-2}$), (E) mean squared shear (s$^{-2}$) (red) and squared buoyancy frequency (s$^{-2}$) (blue) and (F) gradient Richardson number at MR01. (A) and (D) are plotted with respect to height above the bottom and velocity vectors are 6-hour low-passed. Dashed vertical lines denote Storm Desmond.

Averaged over a complete tidal cycle, volume fluxes highlighted that net transport was predominantly seaward during the deployment period (negative values represent outward volume fluxes, Fig 8A). A mean net volume flux of -15.8 m$^3$ s$^{-1}$ was estimated during December, with volume fluxes of up to -80 m$^3$ s$^{-1}$ estimated during Desmond. For comparison, a volume flux estimate made using an ADCP over summer 2010 had a mean value of—3 m$^3$ s$^{-1}$ [38]. Positive volume fluxes (into Furnace) were largely attenuated during early December with inward volume transport occurring during spring tides and up-estuary winds later in the month (Fig 8B). However larger outflow volumes for the remainder of December and January ensured that volume transport was predominantly seaward with only occasional, short-lived periods of net positive fluxes (Fig 8A and 8B).

The tidally-averaged total salt flux was generally negative (seaward) for most of early December with no net positive influx until spring tides just prior to 18$^{th}$ December (black line, Fig 8C). Following this point, the salt flux showed a pronounced spring-neap oscillation with net influx during springs and outflux during neaps, which was most apparent when the salt flux was low-passed to remove oscillations with periods smaller than 5 days (red line, Fig 8C).

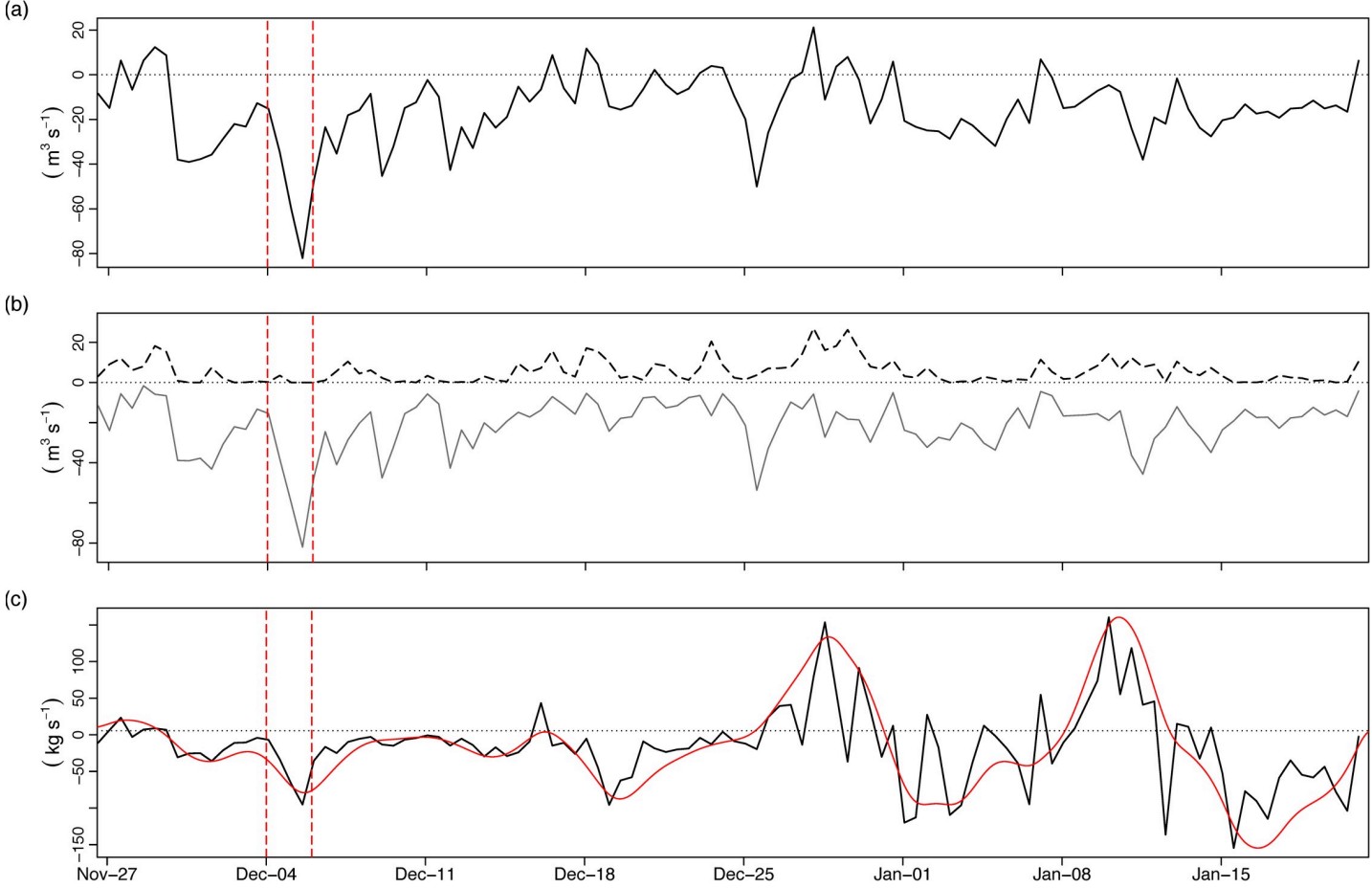

**Fig 8. Tidally-averaged volume and salt fluxes in/out of Lough Furnace.** (A) net volume flux (m$^3$ s$^{-1}$) in/out of Furnace inner basin through MR01 (positive values indicate flux into Furnace; negative values outward flux), (B) partitioned volume fluxes into positive (dashed line) and negative (solid line) components, (C) salt flux (kg s$^{-1}$) in (positive) and out (negative) (red line is 5-day low-passed). Dashed vertical lines denote Storm Desmond.

Of particular importance then was that at peak flooding (4–6 December), salt export increased despite only small quantities of residual salt in the bottom of the channel from the prior spring tides (Fig 7B); this implies that the steady-state salt balance was modified and there was a net loss of salt from the inner Furnace basin during the flood period.

## Discussion

A number of recent reports suggest that the intensity of extreme rainfall and storm events is expected to increase in response to increases in atmospheric heat and moisture [5,6,56]. Thus, documenting how different aquatic ecosystems react to the varying magnitude and timing of episodic events is crucial, as biogeochemical processes and ecological communities may be perturbed. This study has quantified the cascading impacts from successive heavy rainfall events during a single winter on a catchment, its freshwater lake and its downstream estuarine waters, with an emphasis on one exceptional event, the extratropical North Atlantic cyclone Storm Desmond. Most importantly, it has shown how synchronicity between storm-specific forcing factors (precipitation intensity, wind speeds and onshore wind direction) and anteced-ent local conditions (pre-saturated catchment ground and high water levels from prior rainfall) can amplify environmental impacts. Of particular note was how the magnitude of storm

impacts varied between the linked aquatic ecosystems, despite their spatial proximity. Results highlight the relative contribution of conditions associated with large-scale atmospheric variability (e.g. a high-moisture winter storm originating in the North Atlantic) and local factors (e.g. isothermal, winter-mixed freshwater lake versus saline stratified estuarine system) to the nature of the environmental or ecosystem disturbance experienced.

## Storm impacts on catchment and freshwater lake habitats

Analysis of the rain gauge data during Storm Desmond showed that the rainfall was extremely heavy, continuous and spatially distributed throughout the catchment. This was in contrast to an extreme precipitation event documented in this catchment previously, when a much shorter and intensive rainfall event exhibited a notable asymmetry and led to particularly destructive flooding on the eastern side of the catchment [17]. In Lough Feeagh, the most notable shift was the sudden increase in surface water acidity following Desmond. This was likely a consequence of the mobilisation of organic matter from the predominantly peatland catchment area [37] with a potential contribution from the direct rainfall over the lake surface. Whilst Feeagh is normally slightly acidic, post-flood values were noticeably lower than the annual mean pH (6.1 compared to 6.6). River plumes resulting from major flood events have been shown to cause decreases in lake pH relative to non-plume waters, with this effect related primarily to the soil-type surrounding the river [57,58]. These results are applicable to lake systems situated in other upland peat catchments around Ireland and the United Kingdom, that are likely to experience more of these high intensity rainfall events which are projected for these regions [35,36,59]. Significant increases in post-flood turbidity and decreased water clarity have been documented for other lakes [11,20]. Biological consequences of increased riverine input may include increased respiration [17,19,60] whilst decreased water clarity following storms may substantially decrease lake primary productivity [61]. However, as the event described here occurred in winter, when primary productivity was low, and changes in water clarity, inputs of organic material and even loss of plankton communities to bulk advection by flood waters are less likely to cause major ecological change [12,17]. Monthly measurements of water colour, total nitrogen and total phosphorus in our study did not deviate noticeably from usual winter values.

Apart from biogeochemical changes, Feeagh also underwent net warming during the storm due to positive (downward) heat fluxes (Figs 3D and 4). Our results represent a rare quantification of the direct transfer of properties from an 'atmospheric river' [35] directly to surface and interior lake waters. The magnitude of warming was small however and heat was rapidly redistributed throughout the water column. Thus, the winter setting was again important in terms of how effectively air-water energy transfer was circulated to interior waters, as Feeagh is fully mixed and isothermal by December. In contrast, an extreme flood event during summer 2009 when Feeagh was stratified had more drastic impacts on water column hydrodynamics, causing a decrease in stratification stability and deepening the epilimnion [17].

## Storm impacts on estuarine and coastal habitats

The largest storm impacts occurred on the downstream Lough Furnace. Here, storm event timing had less impact hydrodynamically, as this system is stratified year-round [38]. Prior to Desmond, heavy rainfall throughout November had caused river discharge to increase the thickness of the freshwater surface layer, resulting in a large salinity (*viz.* density) gradient with depth and a doubling in stratification strength. The major consequence of a deepening halocline is that it reinforces stagnant, anoxic conditions in the bottom basin water. Whilst Furnace does not overturn vertically, occasional lateral inflows of dense, oxygenated coastal

water can ventilate the bottom anoxic water [38]. Similar deep-water ventilation processes are documented for stratified coastal basins such as fjords and sea-lochs [62] and play a fundamental role in oxygen dynamics (e.g. [63]), microbial activity (e.g. [64]) and primary production (e.g. [65]). In Furnace, deep-water ventilation has been observed to occur only following prolonged dry periods with little river flow, allowing tidal inflows of coastal water to reach the bottom of the inner basin. The large freshwater plume that occurred during winter 2015 blocked communication between the deep basin water and external coastal water, the only potential supplier of oxygenated replacement water. Initially the surface freshwater layer occupied the full depth extent of the connecting channel inhibiting any landward inflows.

Consequently, this absence of stratification in the connecting channel leading up to and during Storm Desmond minimised shear, particularly below the wind-influenced surface layer (Fig 7D). Subsidence of the initial flood and onset of spring tides gave rise to an evolving and highly dynamic flow environment. Tidal currents and strong winds generated intense shear near the surface and bottom, supported by the restratification, with indication of turbulent mixing in the entrance channel (Fig 7F). However the large amount of residual freshwater and successive spring tides meant that increasingly strong stratification ultimately suppressed large-scale turbulence and gave way to a stratified exchange flow regime. Flow structure and turbulence in this section is noteworthy, as it sets the composition of fluid entering and leaving Furnace. This has implications for exchange of resident interior water [38] and influxes of nutrients or planktonic seed populations [66] and also establishes the initial density of the estuarine outflow, which can influence far field plume dynamics in the coastal ocean [67]. The development of stratified flow resulted in the dilution of incoming tidal water as it met largely unmixed low salinity surface outflow in the shallow area between MR01 and MR02 (Figs 1 and 7B); this resulted in only intermediate depth water replacement in the main Furnace basin in the months following the flood event.

More broadly, modified freshwater runoff climatology has been implicated in exacerbating dissolved oxygen depletion in estuarine and coastal systems globally [68]. This is largely due to the mechanism explored here: intensified vertical stratification and isolation of deeper waters from lateral inflows. Additionally, rivers may cause nutrient enrichment which stimulates primary productivity or directly transport organic matter to coastal zones, which both heighten respiration rates and increase oxygen demand [26]. At the largest scale, bottom oxygen dynamics in the Baltic Sea are generally controlled by deep water inflows from the contiguous North Sea; changes in precipitation intensity over North-West Europe in recent decades has been implicated in a reduction in ventilation frequency due to large freshwater outflows attenuating dense inflows [69]. In semi-enclosed coastal systems similar to Furnace with restricted connections to adjacent coastal areas, large seaward river flows can inhibit ventilation of deep waters (e.g. [70,71]). Thus, a very likely consequence of changing patterns of precipitation intensity, especially the kind of extreme flood events recounted here, could significantly reinforce stagnant bottom water conditions in deep estuarine basins, promoting anoxia.

In several studies of stratified basins during storms, upwelling of sub-pycnocline water in nearshore areas was documented [15,18,72] and in general upwelling plays a pivotal role in generating turbulent flux pathways around lateral boundaries [73,74]. Partial upwellings occurred in Furnace during individual storm events throughout December, with the stable stratification preventing complete surfacing of lower layer water (Fig 6C). Pycnocline destabilisation results in energy transfer to the internal wave field, with the ratio of h/H and range of inverse Wb in Furnace during December implying that most of the resulting standing waves were linearly damped [53]. In systems with deep anoxia, internal seiche-related upwelling can have added significance, exposing nearshore areas to drastically fluctuating oxygen levels [32,53,75].

## Catchment-to-coast volume fluxes

A ratings curve developed for the Mill Race and Salmon Leap rivers (Fig 1) provided a useful reference for the anticipated freshwater outflow volumes from the Burrishoole catchment during December (although it should be noted that the ratings curve relationship almost certainly breaks down for the extreme flows that occurred during Desmond). The mean river discharge into Furnace during the month of December was 13.7 $m^3$ $s^{-1}$. This compared reasonably well to the mean volume flux estimated from the ADCP of—15.8 $m^3$ $s^{-1}$, which also would have integrated volumetric fluxes from direct precipitation over Furnace as well as additional surface and groundwater inflows from the surrounding land area. Thus, using the ADCP estimates, the mean volumetric flux leaving the Burrishoole system during Storm Desmond (4–6 December) was (O)43 $m^3$ $s^{-1}$, with instantaneous fluxes potentially as high as (O)88 $m^3$ $s^{-1}$ at peak flooding. Based on the Feeagh lake level records, this was most likely the highest volume of water leaving Burrishoole since 1976 when records began (Fig 2). Mean annual discharge from Feeagh into Furnace is only 3–5 $m^3$ $s^{-1}$ [38]. Furthermore, the total volume of rainfall estimated from analysis of the catchment rain gauges over the 72 hour period (4–6 December) gave a volumetric rainfall flux of 46–54 $m^3$ $s^{-1}$, which again compares reasonably well to the ADCP estimate. The mismatch may be attributed to measurement error and calculation uncertainty attributed to each method and also that not all of the rainfall would have instantaneously exited the system (for example Feeagh lake level took ~10 days to fall back to typical mean December levels after the 6th December (Fig 2)).

Using the average volume flux of 43 $m^3$ $s^{-1}$ as a representative transport of water through the catchment during Desmond, the renewal time of the Feeagh lake basin ($T_R = Vol/Q$) was reduced to 15.8 days. The average renewal time of Feeagh is ~170 days; a previous extreme flood event during summer 2009 indicated that the renewal time of Feeagh also reduced to (O) 15 days [17]. However a major difference was that peak flow rates used to estimate this reduced renewal time during the summer flood lasted only ~3 hours, whereas our estimate for the Desmond volume flux is representative of a 3 day period, meaning that at least 20% of the volume of fully-mixed Feeagh was displaced over the 4–6 December, if a 100% exchange efficiency is assumed.

These values provide a first order estimate of the quantity of freshwater that ultimately reached Clew Bay and the coastal ocean from one single watershed in the aftermath of Desmond. Considering additional sources of river discharge around Clew Bay, it is likely that such elevated freshwater discharge would have affected stratification dynamics in the short term and may have delivered large quantities of terrestrial-source water properties over a very short time span (e.g. [27,28]). Numerical modelling of the impact of such extreme rainfall and flood events on the oceanography and biogeochemistry of Clew Bay could offer insight into how large coastal embayments respond to such climate-related extreme events.

## Storm impacts on estuarine salt balance

In estuarine basins, the salinity budget is maintained by the competing influences of seaward and landward salt transport and depends on tidal, wind and freshwater flow [76]. In stratified estuaries such as Furnace, spring-neap oscillations in salt transport are often the dominant factor in determining the longer-term salt balance (e.g. [29]). For example, in the inner basin of Furnace, steady-state conditions in relation to salt and volume are on average achieved over a complete spring-neap cycle [77]. However large pulses of freshwater may disrupt the longer-term salt balance, especially in lagoonal estuaries with restricted horizontal connection to oceanic water. The Furnace salt balance was modified during Desmond with net export of salt which was not replenished immediately by spring tides. Perhaps most importantly, this salt

flux did not appear to be imported salt from the prior spring tides, as the surface and bottom salinities at MR01 were almost zero prior to Desmond yet both increased slightly during the middle of Desmond (Fig 7B). To assess whether this exported salt came from sub-halocline water in the deep inner basin, we invoked some practical assumptions about the mean volumetric flux during Desmond ($Q_{mean} \approx 43$ m$^3$ s$^{-1}$), the salinity of the water above the halocline at the mouth of the basin ($S_1 \approx 0$, recorded at MR01) and salinity below the halocline ($S_2 \approx 16$; Fig 5A), and estimated an entrainment flux ($Q_{We}$, m$^{-3}$ s$^{-1}$) of sub-halocline water by the outflowing plume as $Q_{We} = Q_{mean}(S_1/S_2 - S_1)$. This provided a mean estimate of $Q_{We}$ during the flood of 2.8 m$^3$ s$^{-1}$. Using the approximate basin area at the depth of the halocline gives a mean entrainment velocity of 5 x 10$^{-6}$ m s$^{-1}$. Thus for $S_2 \approx 16$, a simplistic estimate of mean upward salt flux across the halocline gives 45 kg s$^{-1}$; over the 4–6 December the mean salt flux through the entranceway based on Eq 10 was– 42 kg s$^{-1}$ (although the large range of values in the salt flux timeseries must be acknowledged, with max values of– 90 kg s$^{-1}$ at peak flow (Fig 8C)). Thus we tentatively conclude that salt loss from Furnace during the peak of the freshwater flood was from older resident salt reserves located beneath the halocline and below sill depth. This is an important dynamic, as in topographically constrained estuaries like Furnace, the deep basin water below sill depth is normally detached from the wind, estuarine and tidal circulation that takes place above the sill (e.g. [62]). Large flood events may be capable of freshening semi-enclosed coastal basins such as Furnace not only by limiting salt import during flood tides but also by exporting resident salt through entrainment.

## Conclusions

Here we have quantified the impacts from a succession of heavy rainfall events that spanned a catchment-lake-estuarine system and shown that, in large part due to the timing (winter) and event conditions (rain and wind), the coastal estuarine system underwent the most significant changes. The timing of such events over the annual cycle is less important for estuarine systems in temperate climates and in some respects winter floods may have larger impacts, as extreme rainfall is superposed on higher seasonal freshwater input. A logical next phase to the results presented here would be to quantify how additional biogeochemical and biological parameters not monitored during the flood would respond, as well as tracking plume impacts further offshore. One potential approach toward this would be forcing coupled physical-biogeochemical models using volume and air-sea fluxes quantified here as boundary conditions.

## Supporting information

**S1 Fig. Cross-sectional profile of entranceway into Furnace inner basin at MR01.** Shown is the instrumentation configuration of this mooring (acoustic doppler current profiler and temperature-conductivity sensor (bottom triangle); temperature-conductivity sensor (surface triangle)). Black lines indicate borders of 4 regions used to compute volume and salt fluxes through the entranceway.
(EPS)

## Acknowledgments

The long-term environmental monitoring programme in Burrishoole is facilitated by the technical staff of the Marine Institute Newport (Joseph Cooney, Michael Murphy, Pat Hughes, Pat Nixon, Davy Sweeney) and Martin Rouen (Lakeland Instrumentation Ltd.). The work on Lough Furnace was only possible thanks to field and technical support from Sheena Fennell (Earth and Ocean Science Department, NUI Galway).

## Author Contributions

**Conceptualization:** Seán Kelly, Brian Doyle, Elvira de Eyto.

**Data curation:** Seán Kelly, Elvira de Eyto, Mary Dillane.

**Formal analysis:** Seán Kelly, Brian Doyle.

**Funding acquisition:** Phil McGinnity, Martin White, Eleanor Jennings.

**Investigation:** Seán Kelly, Elvira de Eyto, Mary Dillane.

**Methodology:** Martin White.

**Project administration:** Phil McGinnity, Eleanor Jennings.

**Supervision:** Russell Poole, Martin White, Eleanor Jennings.

**Visualization:** Seán Kelly.

**Writing – original draft:** Seán Kelly.

**Writing – review & editing:** Seán Kelly, Brian Doyle, Elvira de Eyto, Mary Dillane, Phil McGinnity, Russell Poole, Martin White, Eleanor Jennings.

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
