## [Decision Letter · Decision Letter 0]

6 May 2020

PONE-D-19-34756

Impacts of a record-breaking storm on physical and biogeochemical regimes along a catchment-to-coast continuum

PLOS ONE

Dear Dr. Kelly,

Thank you for submitting your manuscript to PLOS ONE. After careful consideration, we feel that it has merit but does not fully meet PLOS ONE’s publication criteria as it currently stands. Therefore, we invite you to submit a revised version of the manuscript that addresses the points raised during the review process.

We would appreciate receiving your revised manuscript by Jun 20 2020 11:59PM. To enhance the reproducibility of your results, we recommend that if applicable you deposit your laboratory protocols in protocols.io, where a protocol can be assigned its own identifier (DOI) such that it can be cited independently in the future. For instructions see: http://journals.plos.org/plosone/s/submission-guidelines#loc-laboratory-protocols

We look forward to receiving your revised manuscript.

Kind regards,

A. P. Dimri, Doctorate

Academic Editor

PLOS ONE

Journal Requirements:

2. We note that Figure 1 in your submission contains a map image which may be copyrighted. All PLOS content is published under the Creative Commons Attribution License (CC BY 4.0), which means that the manuscript, images, and Supporting Information files will be freely available online, and any third party is permitted to access, download, copy, distribute, and use these materials in any way, even commercially, with proper attribution. For these reasons, we cannot publish previously copyrighted maps or satellite images created using proprietary data, such as Google software (Google Maps, Street View, and Earth). For more information, see our copyright guidelines: http://journals.plos.org/plosone/s/licenses-and-copyright.

a).    You may seek permission from the original copyright holder of Figure(s) [#] to publish the content specifically under the CC BY 4.0 license.

Reviewers' comments:

Reviewer's Responses to Questions

**Comments to the Author**

1. Is the manuscript technically sound, and do the data support the conclusions?

Reviewer #1: Yes

2. Has the statistical analysis been performed appropriately and rigorously? 

Reviewer #1: Yes

3. Have the authors made all data underlying the findings in their manuscript fully available?

Reviewer #1: Yes

4. Is the manuscript presented in an intelligible fashion and written in standard English?

Reviewer #1: Yes

5. Review Comments to the Author

Reviewer #1: In this work, the authors investigated the impact of extreme weather event (Storm Desmond) on a catchment-lake-estuarine system (Burrishoole system on the Atlantic coast of Ireland). By the application of a network of high-frequency in-situ sensors, the authors presented how an unprecedented period of rainfall, high wind speeds, and high winter air temperatures affect the coastal systems. In detail, the authors have applied a suite of parameters including: (1) storm effects on dynamical transfer processes of heat and momentum (Qshf); (2) mechanical energy transfer from the wind to the water (E10); (3) buoyancy at Furnace (Bf); (4) full water column stability (Schmidt stability index, Ss), which was further interpreted as total lake heat content at Feeagh (Htot) and Wedderburn number (Wb) at Furnace; (5) dynamic of stability of flow (Richardson number, Rig); and (6) net salt flux (Sf) to quantitatively assess the impact of storm event. In the lake, the rise in water level is accompanied by an increase in turbidity and a decrease in pH at surface during and after storm. In addition, due to the warm and moist above-lake air mass, total lake heat content (Htot) slightly increased, despite an overall low value throughout winter season. In the downstream, the stratification was intensified by the introduction of freshwater (Schmidt stability doubled), and only partial upwelling was detected (Wb value in the range from 1 to 10) after the storm based on automated profiler data. However, the stratification was totally absent and the strong seaward flow occupied the full extent of the channel depth during Desmond. Coincident with the seaward flow, salt export also increased, with a maximum of ca. 90 kg s-1 at peak flow.

Overall, Desmond storm was likely to cause less major ecological changes in the catchment and lake systems, mainly due to the timing of the storm (winter, low primary productivity). On the other hand, the largest storm impacts occurred on the downstream estuarine and coastal systems. Despite the absence of stratification during the storm, the large amount of residual freshwater eventually resulted in an even stronger stratification post storm. This intensified vertical stratification, together with nutrient rich and organic rich river water, was likely to cause exacerbating dissolved oxygen depletion in estuarine and coastal systems, and could have global impacts.

This study has provided a unique angel to investigate the impact of storms on a catchment-to-coast continuum, and presented the results of an original research with a large amount of solid data. Experiments design, and specifically how each parameter was calculated, was described in sufficient details. The conclusions drawn in this work was clear and was well supported by the data. I generally agree the publication of this work, as long as the authors can take care of several minor comments listed here:

The title of this work is the impact of a storm on physical and biogeochemical regimes. This manuscript has provided very solid results on the physical part, but only briefly mentioned biogeochemistry throughout the work (e.g., pH, DO). In the discussion session, the authors did talk about the impact of the storm on the lake system and the downstream coastal system, but the discussions were more like speculation without support of actual data. The manuscript will be strengthened if the authors could include some data about how the nutrients regime changed in the lead up to and immediate aftermath of storm Desmond in both the lake and the downstream coastal regions.

Some specific minor comments are:

Line 96: There is a typo. It should be “between” not “0between”.

Line 473: Should be “(C) salt flux” not “(B) salt flux”.

6. PLOS authors have the option to publish the peer review history of their article (what does this mean?). If published, this will include your full peer review and any attached files.

Reviewer #1: No

---

## [Author Response · Author response to Decision Letter 0]

12 May 2020

We sincerely thank the reviewer for taking the time to evaluate our manuscript and providing positive feedback in relation to the study. We have incorporated changes to the manuscript and here provide a response to each of the reviewer’s comments.

5. Review Comments to the Author

"This study has provided a unique angel to investigate the impact of storms on a catchment-to-coast continuum, and presented the results of an original research with a large amount of solid data. Experiments design, and specifically how each parameter was calculated, was described in sufficient details. The conclusions drawn in this work was clear and was well supported by the data. I generally agree the publication of this work, as long as the authors can take care of several minor comments listed here:"

Thank you very much for the encouraging feedback and your views on the overall quality of the study

"The title of this work is the impact of a storm on physical and biogeochemical regimes. This manuscript has provided very solid results on the physical part, but only briefly mentioned biogeochemistry throughout the work (e.g., pH, DO). In the discussion session, the authors did talk about the impact of the storm on the lake system and the downstream coastal system, but the discussions were more like speculation without support of actual data. The manuscript will be strengthened if the authors could include some data about how the nutrients regime changed in the lead up to and immediate aftermath of storm Desmond in both the lake and the downstream coastal regions."

We agree that the primary results of the study did focus more on the physical aspects, although this was largely related to the physical impacts being much more profound compared with the biogeochemical changes primarily due to the timing of the storm (winter when productivity is low). We had envisaged that this in and of itself is a useful finding.

We do include some analysis of pH and dissolved oxygen as the reviewer notes and did show how pH dropped in the freshwater lake immediately post-storm and also the implications of large river flows for reinforcing stratified conditions that promote deoxygenation in the estuarine zone. We also showed how turbidity increased in the aftermath of the storm (Fig. 3B ; line 359) and a portion of the results and discussion highlight the reduction in salinity in the Lough Furnace estuarine basin (e.g. lines 474-481) and overall freshening effect of such a storm on coastal systems. For example the section on storm impacts on estuarine salt balance in the discussion (beginning lines 616)

Chlorophyll a is monitored on both systems but did not show a significant change given the winter setting. We reference how the timing of this storm was important in minimising some biogeochemical changes and compare it directly to the large chlorophyll a changes observed during a summer storm in the same system (reference 17 and Lines 514-517).

Finally, nitrogen, phosphorus and water colour are analysed from water samples taken on both Feeagh and Furnace but only at a monthly frequency so regrettably we were unable to definitively assess how the nutrient status changed in the immediate lead-up to and aftermath of the storm. However, we do note that these monthly values did not differ from historical values (Line 521-522), again showing that for the freshwater lake at least biogeochemical changes were small compared to big chemical changes in salt and oxygen in the estuary. 

Some specific minor comments are:

"Line 96: There is a typo. It should be “between” not “0between”."

Corrected

"Line 473: Should be “(C) salt flux” not “(B) salt flux”."

Corrected

Additional journal requirements

"1. Please ensure that your manuscript meets PLOS ONE's style requirements, including those for file naming."

We have edited aspects of the manuscript style format to meet the requirements. We have also re-uploaded each of the files using the correct naming format (e.g. Fig 1.eps is now Fig1.eps)

"2. We note that Figure 1 in your submission contains a map image which may be copyrighted. 

We require you to either (a) present written permission from the copyright holder to publish these figures specifically under the CC BY 4.0 license, or (b) remove the figures from your submission:"

We have supplied a replacement Figure 1. The basemap background map layer in the original figure, which contained the copyrighted material, has been removed for the replacement figure.

Updated request for revised map source material on 12/05/2020:

"1. Thank you for updating the basemap of Fig. 1. To ensure that this figure is now compliant with CC BY 4.0 (https://creativecommons.org/licenses/by/4.0/), we ask that you provide us with source information for the basemap, the data points, and the data overlaid on the map."

In the revised Fig. 1 the following data sources where used:

1. Lake shapefiles – from Environmental Protection Agency and available here: http://gis.epa.ie/geonetwork/srv/eng/catalog.search#/metadata/c4040e19-38ec-4120-a588-8cd01ac94a9c. Data is for public use under Creative Commons CC-By 4.0.

2. River shapefiles – from Environmental Protection Agency and available here: http://gis.epa.ie/geonetwork/srv/eng/catalog.search#/metadata/c4043e19-38ec-4120-a588-8cd01ac94a9c. Data is for public use under Creative Commons CC-By 4.0.

3. Data points (coloured, labelled markers on the map) – these are the location of instruments deployed by the authors of this study and have been marked according to GPS coordinates recorded during each deployment.

4. The original basemap, which had the copyrighted material, has been removed for the revised Fig. 1. The current revised Fig. 1 does not contain any basemap and we thus we cannot provide any source information.

5. We have updated the caption of Fig.1 to provide the general source information for the lake and river shapefiles listed in point 1 (http://gis.epa.ie/GetData/Download).

---

## [Editor Report · Decision Letter 1]

26 Jun 2020

Impacts of a record-breaking storm on physical and biogeochemical regimes along a catchment-to-coast continuum

PONE-D-19-34756R1

Dear Dr. Kelly,

We’re pleased to inform you that your manuscript has been judged scientifically suitable for publication and will be formally accepted for publication once it meets all outstanding technical requirements.

Kind regards,

A. P. Dimri, Doctorate

Academic Editor

PLOS ONE
---

## [Editor Report · Acceptance letter]

30 Jun 2020

PONE-D-19-34756R1 

Impacts of a record-breaking storm on physical and biogeochemical regimes along a catchment-to-coast continuum 

Dear Dr. Kelly:

I'm pleased to inform you that your manuscript has been deemed suitable for publication in PLOS ONE. Congratulations! Your manuscript is now with our production department. 

Kind regards, 

on behalf of

Dr. A. P. Dimri 

Academic Editor

PLOS ONE